# Novel SH-SAW Biosensors for Ultra-Fast Recognition of Growth Factors

**DOI:** 10.3390/bios12010017

**Published:** 2021-12-30

**Authors:** Daniel Matatagui, Ágatha Bastida, M. Carmen Horrillo

**Affiliations:** 1Tecnología de Sensores Avanzados (SENSAVAN), Instituto de Tecnologías Físicas y de la Información (ITEFI), Consejo Superior de Investigaciones Científicas (CSIC), 28006 Madrid, Spain; d.m@csic.es; 2Instituto de Química Orgánica General (IQOG), Consejo Superior de Investigaciones Científicas (CSIC), 28006 Madrid, Spain

**Keywords:** growth factor, fast detection, SH-SAW, biosensors, microfluidics

## Abstract

In this study, we investigated a label-free time efficient biosensor to recognize growth factors (GF) in real time, which are of gran interesting in the regulation of cell division and tissue proliferation. The sensor is based on a system of shear horizontal surface acoustic wave (SH-SAW) immunosensor combined with a microfluidic chip, which detects GF samples in a dynamic mode. In order to prove this method, to our knowledge not previously used for this type of compounds, two different GFs were tested by two immunoreactions: neurotrophin-3 and fibroblast growth factor-2 using its polyclonal antibodies. GF detection was conducted via an enhanced sequential workflow to improve total test time of the immunoassay, which shows that this type of biosensor is a very promising method for ultra-fast recognition of these biomolecules due to its great advantages: portability, simplicity of use, reusability, low cost, and detection within a relatively short period of time. Finally, the biosensor is able to detect FGF-2 growth factor in a concentration wide range, from 1–25 µg/mL, for a total test time of ~15 min with a LOD of 130 ng/mL.

## 1. Introduction

Growth factors (GFs) are biomolecules which are involved in a variety of cellular processes, such as proliferation, anti-apoptosis, drug resistance, and angiogenesis [1]. Dysregulated of GFs signaling causes human diseases, such as breast, gastric, and lung cancer [2]. We selected two GFs (NT-3 and FGF-2) in this study due to the great importance in medicine, since they intervene in many biological processes related to diseases [3]. Neurotrophin-3 is a protein encoded by the NTF3 gene. This protein was the third neurotrophin discovered [4], being a neurotrophic factor with activity in certain neurons of the peripheral and central nervous system (CNS) helping the survival and differentiation of existing neurons [5]. The expression of NT-3 starts with the onset of neurogenesis and continues in adult life [6]. These proteins exert their effects through interactions with neuron cell surface receptors, such as specific Trk receptors (receptor tyrosine kinase, TrkA, TrkB, and TrkC) with high-affinity and p75 receptor with lower-affinity [7]. Neurotrophic factors are able to control synaptic transmission, differentiation, plasticity, and protect the nervous system from different types of damage [8]. On the other hand, the growth factor FGF-2 is a member of a large family of proteins that binds heparin and heparan sulfate which modulate the function of a wide range of cell types. This protein stimulates the growth and development of new blood vessels (angiogenesis) that contribute to the pathogenesis of several diseases (prostate cancer, atherosclerosis), normal wound healing, and tissue development. FGF-2 binds to fibroblast growth factor receptors (FGFR1–FGFR4), and they are most commonly promiscuous mitogens [9].

There are a lot of methods to measure biomolecules, such as sensitive detection of deoxyribonucleic acid [10], grow factors [11], and cells [12]. Many of them have been used to detect them for a long time to the present, such as those based on fluorometry, spectrophotometry, or chemiluminescence [13], localized surface plasmon resonance (LSPR) [14], mass spectrometry [15], optical methods [16], the electronic tongue [17], the voltammetric method [18], and electrochemistry [19]. Almost all of them have showed great disadvantages such as: high costs, long operating times, and use of labeled molecules. On the other hand, well-established technologies, such as enzyme-linked immunosorbent assays (ELISA) and radio-immunoassays (RIA), are not portable and complicated to use for point-of-care aims. In addition, these techniques have large processing, use many reactives, and work with high volumes, being, therefore, slow and expensive techniques.

To solve these important inconvenients, in this work, shear horizontal surface acoustic wave (SH-SAW) immunosensors were used, since they show relevant advantages, such as high sensitivity, low cost, low power, ultra-fast recognition, and real-time monitoring, features of great interest for different fields of application [20], such as environment [21], security, food, and biomedicine [22].

The principle of SH-SAW sensors is that the mass loading and the perturbation on the viscoelastic properties, due to the join of specific biomolecules to the functionalized sensing layer, causes a shift of the frequency or phase which are measured with a frequency counter or network analyzer [23]. However, for biosensing applications in liquid media there is the inconvenient of the attenuation of the vertical compression for Rayleigh waves into the environment liquid [24]. This fact causes an important damping and therefore it is necessary that the acoustic waves have a shear horizontal fluctuation to inhibit it, which consists of particle displacements parallel to the surface and perpendicular to the direction of the wave propagation. SH-SAW devices have advantages in this type of biological applications due to their low acoustic losses in liquid [25] so these sensors have been successfully applied for detecting different biomolecules such as DNA, bacteria, and virus [26].

Electrochemical [27] and capacitance-free labeling sensors [28] have been used to detect grow factors (GF) in blood. To our knowledge, in literature, there are not yet clear developments of SH-SAW as sensors to detect the growth factors, in particular fibroblast growth factor-2 (FGF-2) and neurotrophin-3 (NT-3), in real-time and dynamic mode (antigen–antibody) [29].

Therefore, in the present work, we focused our research in demonstrating the feasibility of applying SH-SAW sensors for detecting these two GFs for concentrations where the cancer processes are produced, given their compatibility with microfluidics systems. Microfabrication and microfluidics processes play a great role in the development of point of care diagnostic test.

Among all the sensors described to date SH-SAW are the ones that present a best technology capable of efficiently developing both fluidic manipulation and biosensing to manufacture microfluidic biosensors [30]. These SH-SAW sensors are rigged up by standard micro- and fabrication techniques. Microchannels are made of plastic materials by replica molding, while the SH-SAWs are built by a comb-like metal structure called interdigital transducers (IDTs), which are prepared on a piezoelectric material [31].

The compatibility of SH-SAWs with different cells has also been carried out [32], as well as their improvement of biomolecules binding to surfaces [33]. Therefore, SAW-based systems are promising candidates for lab-on-a-chip (LoC) development, capable of realizing both biosensing and microfluidics functions on a single chip, which can also operate wirelessly [34].

In this paper, we have detected NT-3 and FGF-2 proteins as suitable biological markers for metastasis using SH-SAW ultra-fast devices with microfluidics systems.

## 2. Materials and Methods

### 2.1. Materials

All chemical products and solvents were of analytical grade. The reagents used in this research are the protein BSA (bovine serum albumin), the buffer TBS (tris-buffered saline), 3-aminopropy-triethoxysilane (APTES, A3648), glutaraldehyde (GA, G7776), toluene, and 2-butyl alcohol which were supplied by Sigma (St. Louis, MO, USA). Rabbit polyclonal antibody FGF-2 (500-P18–50 μg, FGF basic), rabbit polyclonal antibody NT-3 (500-P82–50 μg), FGF-2 and NT-3 growth factors were supplied by Peprotech.

### 2.2. Assembly of the Biosensor (SH-SAW)

The SH-SAW device (size of 30 × 40 × 0.5 mm) was used for this research (see Figure 1). SH-SAW is transmitted on the ST-cut quartz, in a perpendicular way to the x crystallographic axis. The wavelength of 28 μm was generated and detected by interdigital transducers (IDTs). The IDTs were performed through standard lithographic techniques by depositing an aluminum coating of 200 nm by the RF sputtering technique. The structure of a double electrode was duplicated 75 times to make each of the IDTs. The characteristic parameters of the device are as follows. Spacing, among the IDTs, was 225 λ, and the acoustic aperture was 75 λ; the sensitive surface area was 7.4 mm^2^ (taking into account that the microchannel width was 3.4 mm). Finally, a surface acoustic wave of Love type was attained for guiding the SH-SAW in a SiO_2_ coating deposited by PECVD technique. The maximum sensitivity was achieved for a thickness of approximately 3.7 μm of this oxide [35,36].

### 2.3. PDMS Chip and Liquid Cell

To achieve a homogenous flow in the device, Comsol software was used considering the microchannel shape (Figure 1).

A polydimethylsiloxane (PDMS) chip was performed with a SU-8 mold with the microchannel shape. A SU-8 negative photoresist coating (150 μm) was deposited on a silicon wafer; then it was exposed to an optical lithography technique, and finally baked to attain the master. The microstructure of PDMS and the SH-SAW device were joined by pressure. In this way, microchannels with a height of 150 μm were formed. Figure 2 depicts the realization of the mounting of the PDMS chip on the device SH-SAW.

### 2.4. System Temperature Control

A temperature control system was designed for the Love device for liquids, combining (Figure 2):A Peltier device that allows thermostatization of the socket where the Love device is located. Said plinth made of aluminum, due to its great thermal conductivity, allows the temperature of the device to be homogeneous, and also, due to its low heat capacity, the Peltier device is able to act on its temperature in a short time;A Platinum resistor (Pt100) that is located in a cavity between the socket and the SH-SAW device, and is capable of measuring temperature with an uncertainty of ±0.1 °C;A PID controller, which every two seconds measures the resistance of the Pt100, processing and storing the temperature data to finally act on the Peltier device through a power source;An aluminum heat sink that is used to cool and heats in short times, and besides promotes heat exchange with the environment;A fan is necessary so that the heat exchange between the heat sink and the environment is faster.

### 2.5. Activation of the SH-SAW Devices

The preparation of the surface of the SH-SAW devices used for these experiments was previously described by our group [35]. Briefly, the oxidation of SiO_2_ surface was performed depositing fresh piranha etch (H_2_SO_4_:H_2_O_2_, 3:1, *v*/*v*) for 5 min, then the device was washed with water and nitrogen drying. Then, the SH-SAW device was functionalized by primary amine groups by immersing in a solution of 20 mM of APTES in toluene for one hour and followed by cleaning in toluene and subsequently in 2-butyl alcohol during 10 min. in each case. The surface activated was treated with glutaraldehyde in water at 20 mM for two hours with shaking, and then washed with water in order to remove all reagents that did not react [37]. When not being used for the experiment, the device was stored at 4 °C.

### 2.6. Experimental Setup

The measurement was carried out with a continuous flow through the microchannel of the SH-SAW sensor. The biological sample was deposited at one end of the channel by using a cone, and at the other end of the microchannel there was another cone through which the solution came out when it was connected to a syring pump (KDScientific 210). The sample was exposed in a continuous flow mode between 10 and 20 μL/min. After being injected, the sample passed through the microtubules from the starting cone to the pump syringes. Unbound biological components were stored in syringes. The pump controls the test flow. SH-SAW sensors are sensitive to temperature changes during testing. Thus, the temperature was kept constant at 30 °C using a Peltier device controlled by a PID system from the PC.

Each of delay lines of the SH-SAW sensor were joined to an amplifier circuit, forming an oscillating system that transmitted the signal to a frequency counter. All the components necessary for the biosensor to work, such as the multimeter that measures the resistance (Pt100) (Keithley 2001, Madrid, Spain), the frequency counter (Agilent 53131A, Madrid, Spain), the power supply that controls the temperature (Peltier) (Agilent E3646A, Madrid, Spain), as well as the acquisition of the results, were controlled in real time by a protocol performed for this particular experiment (GPIB) [38].

### 2.7. Detection of the Biomolecules (GFs)

The device was equilibrated with TBS buffer to pH 7.0 at a flow rate of 20 µL/min using a volume of 100 µL. When the frequency and temperature were constant at a flow of 10 µL/min, an antibody solution (anti-FGF-2, 80 µL, at 50 μg/mL, or anti-NT-3, 80 µL, at 50 μg/mL) was passed. Once the decrease in frequency was balanced, the device was washed with 100 µL of TBS buffer, and in some cases the glutaraldehyde groups of the support that had not reacted with the antibody were blocked with a 3% of BSA (80 µL) at a flow of 10 µL/min. Then, the chip was equilibrated with TBS buffer and a solution of the biomolecule of interest was added to detect (FGF-2, 80 μL from 1 to 25 μg/mL, or NT-3, 80 μL to 50 μg/mL), at a flow of 10 µL/min until equilibrium. Figure 3 depicts a schematic of the device sensing the growth factor FGF-2.

## 3. Results and Discussion

The detection of growth factors was performed using gravimetric (mass-sensitive) biosensors, specifically SH-SAW ones, using microfluidic technology.

NT-3 and FGF-2 proteins were selected as growth factors. To detect these proteins, the surface of the SH-SAW device was activated with glutaraldehyde to achieve the union with the anti-growth factors (anti-GF) by the amino groups (–NH_2_) of the lysine or arginine residues. The detection was carried out by the resonance frequency changes of the device due to the increase of mass in its surface. The increment of mass, due to the antibody-antigen affinity, produced a change in the rate of the surface acoustic wave, and therefore a change in the resonant frequency.

The complete detection process of the growth factors, NT-3 and FGF-2, are shown in Figure 4 and Figure 5. First, the activation of the device was carried out as is described in the method section. Once the frequency was stabilized with buffer, anti-NT-3 was incorporated in the cone, observing a frequency decrease of 12.5 kHz due to the binding of the amino groups of the anti-NT-3 with the aldehyde groups anchored on the device surface. Afterward, BSA protein was used to block aldehyde groups not bound to anti-NT3, but there was hardly any change in frequency. Then, buffer was added to stabilize the process and NT-3 protein was incorporated producing a shift of frequency of 6.74 kHz (Figure 4a). The whole experiment was repeated, except that BSA step was skipped, and a frequency shift of 7.59 kHz was obtained for the same concentration of NT-3 (Figure 4b), which is a low difference of a 12%. In this way, it has been seen that the BSA is not essential to block the biosensors before the addition of the GF (Figure 4c), being therefore one of the facts that makes this test faster than ELISA test.

In the case of FGF-2 detection, anti-FGF-2 protein was incorporated in the cone of the system, with a frequency decrease around 12.6 kHz observed due to the binding of amino groups of anti-FGF-2 with the aldehyde groups anchored on the device surface, which was very similar to the case of the anti-NT-3 (both antibodies have similar weight). Afterward, BSA protein was added to block aldehyde groups not bound to the anti-FGF-2 (Figure 5). As it is shown for NT-3 detection, it was not necessary the use of BSA protein either, because hardly any change in frequency was observed.

Finally, after rinsing with buffer to stabilize the frequency, a 25 μg/mL FGF-2 protein concentration was added and was bound with the anti-FGF-2, producing a shift of frequency of 20.91 kHz. To optimize the testing flow rate with this type of devices, ultra-fast recognition assays of FGF-2 at 5 and 10 µL/min were carried out. For 5 µL/min no-saturation was achieved in 5 min.; therefore, it was decided to make the assays with a flow rate of 10 µL/min.

Due to this great response for FGF-2, two lower concentrations of this GF were tested: 2.5 and 1 μg/mL, obtaining a good and discriminated response in a few minutes (Figure 6). The frequency shifts were: 2.89 kHz and 1.22 kHz, respectively. Moreover, the noise was 10 Hz/min; thus, as for 1 μg/mL, when the signal was 232 Hz/min then it would be possible to detect a 130 ng/mL concentration, which would be the limit of detection (LOD).

To obtain an optimized and fast process to detect concentrations of GFs from the sensor responses during the immunoreaction, these ones could be fitted by means of a plateau followed by two phase decays (Figure 7). The R^2^ values were higher than 0.99 for the three concentrations, which reflects a very good agreement with the measurements carried out (Figure 7).

For all cases, the SH-SAW sensor operated in dynamic mode with an adequate flow at 10 μL/min when using micro-channels to detect the biomolecules in a very short time. The dynamic mode promotes the binding between the anti-GF and GF in a fast time and in addition, this sensing system is label free.

## 4. Conclusions

This article describes the detection of GFs, in a very fast and sensitive way and without the need for labeling, using SH-SAW sensors. In addition, these biosensors have the great advantage of observing in real-time the binding interactions of anti-NT-3/NT-3 and anti-FGF-2/FGF-2. BSA is not essential to block the functional groups on the biosensors before the addition of the GF, therefore performing this test in a faster way than the ELISA test. For FGF-2, three concentrations (1, 2.5, and 25 μg/mL) were detected, obtaining a good response and discrimination among them. The LOD for FGF-2 was 130 ng/mL.

Efficient and reusable biosensors have been developed to detect the concentration levels that there are in human blood when processes of metastasis exist. Therefore, the SH-SAW biosensor technology is a hopeful analysis tool for applications in clinical test and it can compete with the already well-established ones but much more complex techniques, such as SPR (surface plasmon resonance) and ELISA (enzyme-linked immunosorbent assay).

## Figures and Tables

**Figure 1 biosensors-12-00017-f001:**
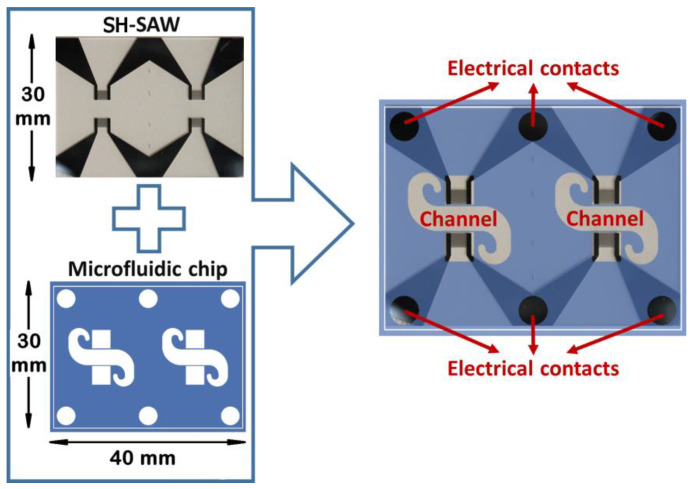
Devices (SH-SAW) plus PDMS microfluidic chip with the microchannel shape.

**Figure 2 biosensors-12-00017-f002:**
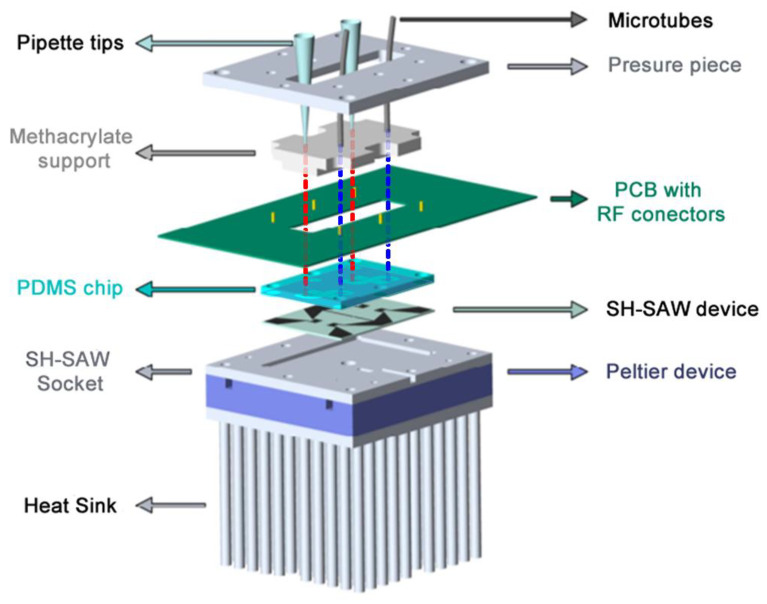
Liquid cell diagram showing the different components.

**Figure 3 biosensors-12-00017-f003:**
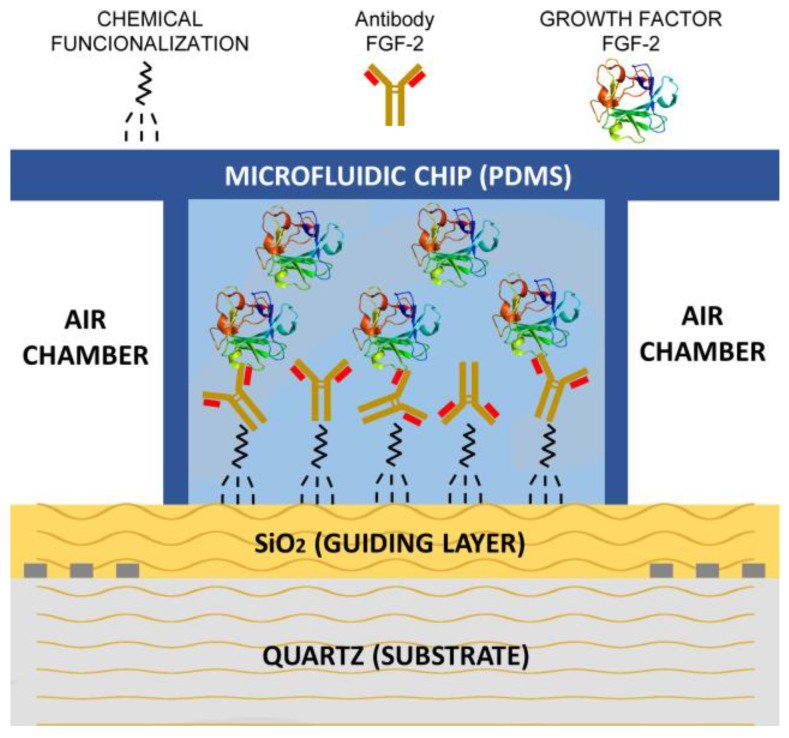
Schematic of the device sensing the growth factor FGF-2.

**Figure 4 biosensors-12-00017-f004:**
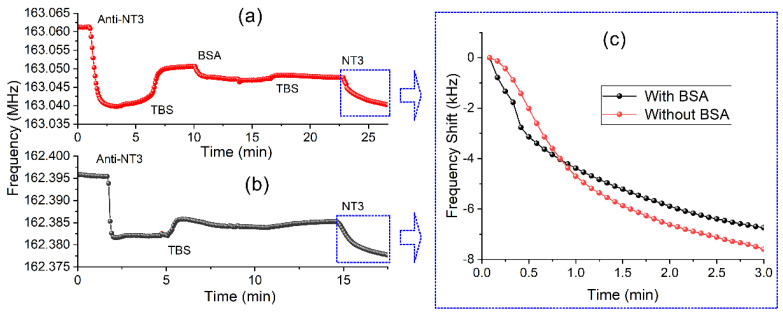
(**a**) Complete sensing process for NT-3 (50 μg/mL) with BSA. (**b**) Complete sensing process for NT-3 without BSA. (**c**) Comparison of frequency shift for both processes.

**Figure 5 biosensors-12-00017-f005:**
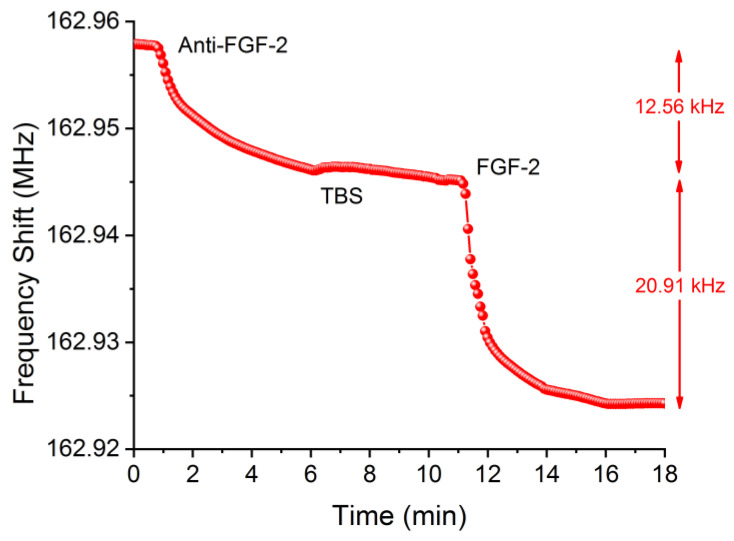
Complete sensing process for FGF-2 (25 μg/mL) without BSA.

**Figure 6 biosensors-12-00017-f006:**
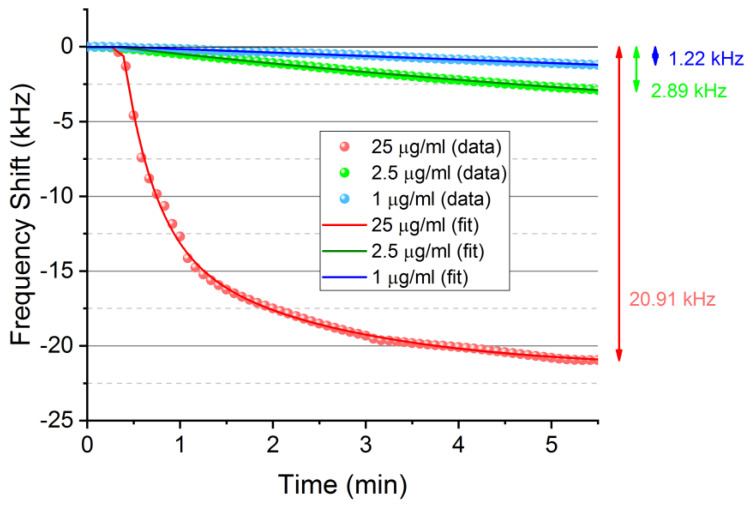
Responses for different concentrations of FGF-2.

**Figure 7 biosensors-12-00017-f007:**
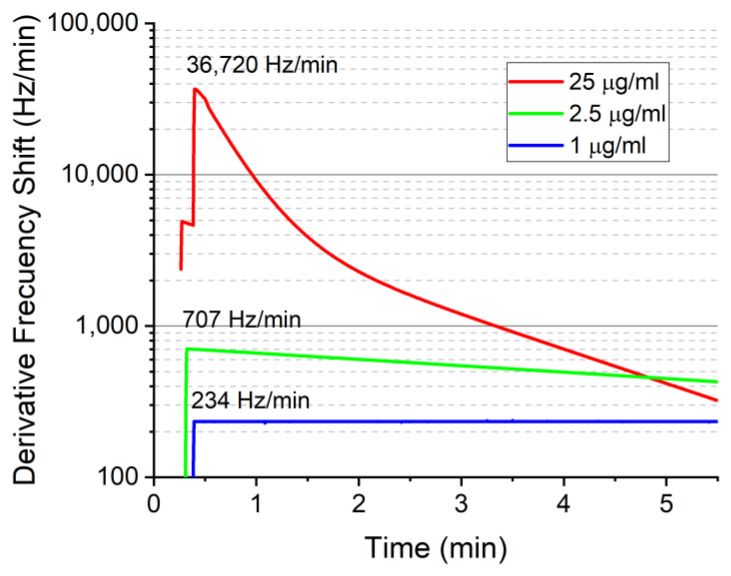
Fitted responses for the three concentrations of FGF-2 detected.

## Data Availability

Not applicable.

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
