# Peer review of "Novel SH-SAW Biosensors for Ultra-Fast Recognition of Growth Factors"

_biosensors, 2021, doi:10.3390/bios12010017_

Round 1

Reviewer 1 Report

Recommendation: Minor revision

The authors reported the applying of an SH-SAW biosensor for dynamic monitoring of growth factors. By testing two different growth factors, including neurotrophin-3 and fibroblast growth factor-2, via an enhanced sequential workflow, the total test time of the immunoassay can be improved. In the end, the authors demonstrated the detection of FGF-2 growth factor in a concentration with wide range (1-25 μg/mL), for a total test time around 15 minutes with SH-SAW sensors. Overall, the topic itself is very interesting, and the work is quite thorough. However, I have a few concerns with some parts of the work which the authors should address before the paper can be accepted.

Major comments:

  • Innovation: Recently, SH-SAW sensors have been extensively investigated in biosensing, such as the detection of DNA, protein and biomolecular. The reviewer can understand that the authors applied the SAW sensors for testing of two new growth factors, however, researchers do have utilized quartz-based SH-SAW sensors for monitoring of growth factors. In terms of innovation, or the claimed novel SH-SAW biosensors in the title, the authors need to convince the reviewer regarding the novelty of this manuscript. Can the authors discuss more regarding the innovation of the sensor design or setup? Is there any difference for testing the two new growth factors with the testing of epidermal growth factor that have been demonstrated with similar quartz-based SH-SAW sensors?

Detailed comments:

  • Line 120: How the authors evaluate the shape of microfluidic channel to achieve homogenous flow with COMSOL software? Is there any reference or simulated results to support this claim?
  • Fig. 1: Scale bar should be added.
  • Fig. 2: There is no microfluidic channel in the illustrated PDMS chip. The connection between pipette tips and PDMS chip is quite confusion.
  • Fig. 3: The color contrast between IDT and substrate can be improved. Now they are both in gray color.
  • Fig. 4b: Why the monitoring time of NT-3 without BSA is much smaller (decreased around 10 minutes) than that with BSA in Fig. 4a?
  • Fig. 5: After injecting the anti-FGF-2 protein, why there is no plateau, such as shown in Fig. 4a, b after injecting the anti-NT-3, during the sensing process?
  • Fig. 6: Can the authors comment on the governing equations or the underlying mechanisms of the fitting curves for different FGF-2 concentrations?
  • Fig. 6-7: The reviewer is curious about the result with a concentration of 5, 10, 15, and 20 μg/mL. Can the authors comment whether they are curves with the shape of 1 μg/mL or 25 μg/mL?

Author Response

Thank you very much for your interesting questions and comments.

Reviewer 2 Report

Comments:

Report on "Novel SH-SAW biosensors for ultra-fast recognition of Growth Factors" by Daniel et al., submitted to Biosensors.

The authors present the label-free time efficient biosensor to recognize growth factors (GFs) in real time, which are of great interesting in the regulation of cell division and tissue proliferation. They claimed with this setup they could achieve a very promising method for ultra-fast recognition of these biomolecules, with great advantages: portability, simplicity of use, reusability, low cost, and detection within a relatively short period of time. However, the following issues should be addressed to strengthen the content and I recommend the publication with major revisions:

  1. How does the optimized testing flow rate (10 ul/min in the paper) is determined? Were there any group experiments results? How does the flow rates affect the testing resolution? Will it greatly break the sensor sensitivity when the flow rates larger than 30 ul/min?
  2. What is the Q factor (quality factor) of this sensor?
  3. For experiments of neurotrophin 3 and fibroblast growth factor-2 recognition, how many repeated experiments were conducted respectively? What are their statistic sd value?
  4. The authors have separately presented the sensor recognition ability of neurotrophin 3 and fibroblast growth factor-2, what about the discernibility for real mixing samples test? What are the sensitivity of the mentioned growth factors existing in the diluted serum for potential clinical application?

Author Response

(The authors gave the same response as above.)

Round 2

Reviewer 2 Report

Agree to publish.